# Factors Differentiating the Level of Functional Fitness in Polish Seniors

**DOI:** 10.3390/ijerph17051699

**Published:** 2020-03-05

**Authors:** Danuta Umiastowska, Joanna Kupczyk

**Affiliations:** 1Faculty of Physical Culture and Health, Institute of Physical Culture Sciences, University of Szczecin, 71-521 Szczecin, Poland; 2Faculty of Tourism and Health Sciences, The Jacob of Paradies University, 66-400 Gorzow Wielkopolski, Poland; jkupczyk@ajp.edu.pl

**Keywords:** functional fitness, Polish seniors, Fullerton Functional Fitness Test

## Abstract

In this study, functional fitness is defined as the level of independence and self-sufficiency of an elderly person, which facilitates leading an independent life, without the need for assistance from other people. A decrease in functional fitness among older adults is influenced by a variety of factors. In addition to changes occurring in the human body in accordance with the laws of ontogenetic development, they can also be analyzed in terms of somatic parameters and the age of the subjects. The aim of this research was to find the factors differentiating the level of functional fitness of older adults. It involved 509 people divided into a group of people involved in regular physical activity and an inactive group. The Fullerton Functional Fitness Test was used to measure the level of physical fitness, and anthropometric measurements (body weight and height) were also performed. The level of functional fitness of Polish seniors was compared with the American standards established by R. E. Rikli and C. J. Jones. The results of our research confirm a higher level of functional fitness in active older adults, both women and men. In this group, BMI—(Body Mass Index is a measure of body fat based on height and weight) showed a greater correlation with shoulder girdle and back muscles strength, complex coordination, balance, agility, and endurance in women than in men. Finally, the surveyed Polish seniors exhibited a higher level of functional fitness than their American peers.

## 1. Introduction

The period of senior adulthood starting at 60–70 years of age is inevitably accompanied by multi-directional changes in motor skills, which are closely related to the lifestyle and state of health in the earlier decades of life [1].

Aging is also accompanied by a reduction in muscle strength, primarily due to a decrease in physical activity and muscle mass. Elderly and senile people are also less capable of adapting to external factors, e.g., to stressful stimuli, and exhibit a slower return to homeostasis. They experience changes in the chemical composition of blood, blood pressure, and body heat. The nervous system also becomes affected, which is manifested by increased emotional lability, weakened memory and concentration, and general mental deterioration [1,2]. There are also the so-called geriatric giants, i.e., immobility, instability, incontinence, impaired intellect/memory, visual and hearing impairment, depression, and falls. A deteriorated functional state indicates an increased need for assistance, either by family or institutions. This has become a particularly important social issue now, with the imminent increase in the proportion of older adults in the population. Fortunately, appropriate preventive measures can and should counteract the premature disability of older adults or aggravation of the existing disabilities.

The role of physical activity in the human life has been emphasized in many research projects indicating the importance of physical activity in the prevention of many diseases, a reduction in the parameters of physiological aging, and an increase in the level of functional fitness in older adults [3,4,5], defined as independence and self-sufficiency of an individual, which allows functioning without the help of others, such as carers or family. This independence requires sufficient strength to get up from bed or an armchair, get in and out of the car, move around in the living environment, do the shopping, and climb stairs. The level of agility and flexibility should be high enough to fasten the seat belt in a car, dress oneself, and to put on one’s shoes. Finally, an appropriate level of complex coordination and balance minimizes the risk of falls, the greatest threat to the elderly. With the right level of endurance, it is possible for senior citizens to be active in their daily lives, go out for a walk, or take advantage of various forms of recreation [6,7]. “By eliminating physical inactivity, life expectancy of the world’s population may be expected to increase by 0.68 years” [7] p.227.

An important element of aging in good health is the ability to function independently, and the foundation of this independence is functional fitness. The phenomenon of loss of independence is a consequence of physiological aging. Its manifestation increases with age and causes the loss of functional independence [8,9,10,11,12].

The development of optimal programs aimed at improving the independence, safety of movement, and broadly understood quality of life has become a challenge for the sciences of health and physical education [13,14,15]. Assessment of physical fitness should be helpful in taking actions that will improve the quality of life of older people [16].

The aim of this research was to determine the factors responsible for differences in the level of functional fitness among older adults.

## 2. Materials and Methods

The research methods used in this work were based on achievement tests using the following research tools:


*Fullerton Functional Fitness Test kit for measuring physical fitness levels [6,7], anthropometric measurements of body weight and height.*


The Fullerton Functional Fitness Test (also known as the Senior Fitness Test (SFT)) was developed by Rikli and Jones at the Lifespan Wellness Clinic at California State University, Fullerton, to measure physical fitness [17,18]. This test is recommended by the International Council on Sports and Physical Education (ICSSPE) as extremely useful for the multidimensional assessment of fitness in older adults. It was published for the first time in 1999 as a field test (i.e., non-laboratory). It is deemed to give the most reliable and complete picture of an individual, as well as variability of functional capabilities among older adults [19].

The testing program included 6 tests measuring the level of physical fitness of seniors (Table 1).

Measurements of height (precision 0.1 cm) and body weight (precision 0.1 kg) were made using a TANITA WB380-H measuring device (TANITA Corporation of America, Inc., Arlington Heights, IL, USA), and BMI was calculated on their basis. The following weight categories were adopted according to the criteria given by the WHO: underweight < 18.5 kg/m^2^; standard 18.5–24.9 kg/m^2^; overweight 25–29.9 kg/m^2^; obese > 30 kg/m^2^ [20].

The study was approved by the Bioethics Committee at the Regional Medical Chamber in Szczecin and involved 516 people (306 women and 210 men) aged 60 to 87 years (mean 69.1 ± 7.18; median—69 years). The mean age for men was 70.9 ± 8.17, median 70, and for women it was 69.1 ± 6.89, median 66. The main characteristics of the subject group are presented in Table 2.

The research was conducted on two groups of older adults. Group 1 included the people who stayed at the rehabilitation and recovery camp at the Rehabilitation and Recreation Centre in Goscim [6], and group 2 included seniors from Mysliborz and Szczecin. In group 1, the study was conducted at the beginning of each camp. Then, an intensive rehabilitation program of 90 min of daily routine was proposed to study participants. It consisted of 30 min of morning exercise and 60 min of general rehabilitation exercises (including exercises to strengthen muscle groups, stability exercises, including exercises on unstable surfaces and stretching). In addition, each participant could engage in other activities, such as Nordic walking, canoeing, or water biking. In group 2, there were women and men who systematically participated in various forms of physical activity carried out as part of the Third Age University curriculum (e.g., Nordic walking, swimming, gymnastics, yoga) and at fitness clubs. The exercises program included two weekly sessions of 60–120 min per week. All participants exercised for at least 1 year. The measurements were made by the authors of the presented article personally in order to reduce the risk of measurement errors, which could occur with a large number of researchers.

Prior to the statistical processing of the collected data, the distribution of the tested variables was assessed using the Shapiro–Wilk test, which showed a normal and close-to-normal distribution. In the analysis of significance of the study results, the Student’s paired *t*-test and the Pearson simple correlation coefficient (r_xy_) were used.

## 3. Results

In the male group, the mean body height was 175.4 cm ± 5.26, median—176, and among women, the mean body height was 162.3 cm ± 4.77, median—163. The mean body weight of men was 82.6 kg ± 12.58, median—82 kg, and among women, the mean weight was 69.4 kg ± 10.23, median—67.

In the male group, the average BMI was 28.1 ± 4.82, and in the female group, the average was 27.4 ± 3.97. The percentage distribution of mean values in BMI categories was as follows: among men, more than half (54%) were overweight; 27% had BMI within the norm; 14% had BMI indicating significant class 1 obesity; and 5% had BMI indicating significant class 2. The situation among women was slightly different, with 37% within the norm, 43% overweight, 10% class 1 obese, 7% class 2 obese, and 3% underweight.

Our comparison of the level of functional fitness among the older adults participating in the improvement and rehabilitation and recovery camp at Goscim (group 1) and the physically active seniors (group 2) showed that the people from group 2, both women and men, showed much higher scores in in 6 tests Fullerton tests. The differences were statistically significant at α = 0.001. In both groups, women were able to reach further in the back scratch and chair sit-and-reach tests than men (Table 3 and Table 4).

The comparison of the results demonstrated by Polish seniors with the only existing standards proposed by Rikli and Jones [18,21] was in favor of our subjects (Table 5 and Table 6). Taking into account the age factor, it can be observed that both Polish women and men achieved moderate results, exceeding the upper limit of the range in the American standards for the following tests: arm curl, chair sit-and-reach, and 8-foot up and go. In the back scratch test, better results were obtained by our group 2 of active Polish seniors. In the 30 s chair stand and 2-min step tests, the average results of Polish respondents reached the upper limit of the standard range in the American research.

The analysis of the relationship between the age of the seniors tested and the level of functional fitness measured by the Fullerton test (Table 7) showed significant relationships for back scratch, 8-foot up and go and 2-min step test in active women (group 2). Among the men, statistically significant relationships were found between groups 1 and 2 in the back scratch, 30 s chair stand in group 1, and chair sit-and-reach in group 2.

Analyzing the relationships between BMI and functional fitness scores (Table 8) in the group of active older adult women (group 2), statistically significant correlations were found for back scratch, 8-foot up and go and 2-min step test (group 2). On the other hand, in men, statistically significant relationships occurred only in the active group (group 2) in relation to back scratch and chair sit-and-reach attempts.

## 4. Discussion

Aging is associated with involuntary processes which lead to the atrophy of muscle tissue (sarcopenia) causing a decrease in strength and endurance. As the decrease in muscle mass usually affects the lower limbs and lower torso, older adults are at increased risk of imbalance and falls [11,12,13,14,15]. Although its scope and pace may differ between individuals, a lower level of functional fitness is associated with the risk of loss of functional independence in everyday life, and so the motor abilities of older adults need to be monitored and evaluated in order to implement appropriate programs and procedures [3,16,17,18]. One of the most important problems that needs to be addressed is the preference for a sedentary lifestyle and reduction of movement to just the most necessary daily activities [19].

According to van Heuvelen and co-authors [22], an increase in physical activity leads to the improvement of functional fitness and therefore increased independence; according to Elphick [23], this can be achieved even in individuals over 90 years of age.

A study by Ignasiak et al. [24,25] on the residents of social welfare homes and the temporary residents of sanitariums showed higher functional fitness scores among the latter, although in two tests (8-foot up and go and 2-min step) they did not reach the American standards proposed by the authors of the Fullerton Test.

Zieliński [19] evaluated the level of functional fitness of 1017 Polish women and observed that they achieved scores below the American standards in four tests, namely: 30 s chair stand test, chair sit-and-reach test, 2-min step test, and 8-foot up and go test.

Król-Zielińska et al. [26] compared the results demonstrated by women and men from the city of Poznań with the American standards and found that in the 60–69-year-old women’s category, the results in the arm curl and 2-min step tests were similar to those of American women, and lower in the 8-foot up and go test, which may indicate lower agility and dynamic balance [26].

The level of functional fitness among the tested Polish seniors (from Szczecin, Mysliborz, and Goscim) was higher than that of their American peers.

Croatian seniors have also achieved results convergent with our subjects in five tests (30 s chair stand test, chair sit-and-reach test, 2-min step test, 8-foot up and go test, arm curl test). This is another group that presents a level of functional fitness higher than American seniors [16].

The comparison of the results achieved by American [27,28,29] and Polish researchers [19,24,25] with the seniors observed in this study shows that active seniors demonstrated a higher level of functional efficiency in all the tests. It can be assumed that this was the result of systematic physical activity.

BMI differentiated upper body strength, complex coordination, balance, agility, and endurance levels in women more often than in men. BMI differentiated the level of lower body flexibility in active men more often than in women. Age differentiated the level of upper body flexibility, complex coordination, balance, agility, and exercise endurance in the groups of physically active older adult women more often than in all the other groups tested.

## 5. Conclusions

The results of our own research confirm a higher level of functional fitness in older adults who exercised on a regular basis, both women and men, compared to the group who did not take part in such activities. Women in this active group showed stronger relationships between the components of individual fitness and somatic parameters (BMI) than men. There was no correlation between the somatic parameters and the level of functional fitness in the group of people on rehabilitation holidays, and between age and the level of functional fitness in women on rehabilitation holidays.

The results also show that regular moderate physical activity better contributes to the overall functional fitness in the elderly than their participation in rehabilitation camps.

## Figures and Tables

**Table 1 ijerph-17-01699-t001:** Fullerton tests of the various parameters of elderly people’s agility.

Course of the Trial	Estimated Parameter
arm curl	upper body strength
chair stand	lower body strength
back scratch	upper body flexibility
chair sit-and-reach	lower body flexibility
8-foot up and go	agility/dynamic balance
2-min step test	aerobic endurance

**Table 2 ijerph-17-01699-t002:** Main characteristics of the study population.

Sex of Respondents	Group 1		Group 2		Total	
	n	%	n	%	n	%
Women	153	59.3	153	59.3	306	59.3
Men	105	40.7	105	40.7	210	40.7
Total	258	52.1	258	47.9	516	100.0

**Table 3 ijerph-17-01699-t003:** Differences in the level of functional fitness among the older female adults.

Course of the Trial	Group 1	Group 2	d	Student’s *t*-Test
arm curl	14.43 (4–28)	19.20 (14–36)	4.77	11.95 **
30 s chair stand	11.24 (1–30)	14.96 (10–32)	3.72	10.21 **
back scratch (cm)	−8.65 (−46–11)	2.40 (−20–19)	11.05	12.26 **
chair sit-and-reach (cm)	3.06 (−38–20)	8.90 (−12–28)	5.84	6.58 **
8-foot up and go (s)	7.66 (3.21–29.01)	5.89 (5.02–19.00)	−1.77	8.41 **
2-min step test (s)	76.02 (11–157)	99.80 (56–219)	23.78	14.22 **

** statistically significant difference at α = 0.001 (source: own research).

**Table 4 ijerph-17-01699-t004:** Differences in the level of functional fitness among the older male adults.

Course of the Trial	Group 1	Group 2	d	Student’s *t*-Test
arm curl	17.71 (8–31)	21.16 (16–40)	3.45	11.21 **
30 s chair stand	12.76 (0–28)	17.00 (15–32)	4.24	11.08 **
back scratch (cm)	−14.93 (−58–10)	−4.30 (−31–18)	10.63	12.34 **
chair sit-and-reach (cm)	1.53 (−32–10)	7.12 (−14–32)	5.59	6.88 **
8-foot up and go (s)	6.86 (3.78–23.70)	5.30 (3.16–19.54)	−1.56	6.37 **
2-min step test (s)	81.88 (13–154)	96.90 (74–202)	15.02	11.06 **

** statistically significant difference at α = 0.001 (source: own research).

**Table 5 ijerph-17-01699-t005:** Comparison of the level of functional fitness of the examined older adult women in different age categories with the American standards.

Course of the Trial	Group	60–64 y.o.**	65–69 y.o.	70–74 y.o.	75–79 y.o.	80–84 y.o.
arm curl	1	18.28	18.45	18.57	15.88	18.41
	2	19.24	20.40	19.90	17.50	19.00
Normal Range of Scores *		13–19	12–18	12–17	11–17	10–16
30 s chair stand	1	15.36	14.33	14.66	12.43	15.37
	2	16.24	16.03	15.90	14.08	16.75
Normal Range of Scores		12–17	11–16	10–15	10–15	9–14
back scratch (cm)	1	−4.75	−7.40	−1.26	−9.07	−0.82
	2	1.25	1.02	0.80	0.69	0.35
Normal Range of Scores		−3.0–1.5	−3.5–1.5	−4.0–1.0	−5.0–0.5	−5.5–0
chair sit-and-reach (cm)	1	6.24	7.76	8.93	4.83	7.66
	2	7.46	8.25	7.80	5.90	6.50
Normal Range of Scores		−0.5–5.0	−0.5–4.5	−1.0–4.0	−1.5–3.5	−2.0–3.0
8-foot up and go (s)	1	6.10	6.64	5.87	8.47	6.40
	2	5.20	5.90	5.10	6.80	6.30
Normal Range of Scores		6.0–4.4	6.4–4.8	7.1–4.9	7.4–5.2	8.7–5.7
2-min step test (s)	1	95.46	90.95	96.56	82.79	94.20
	2	104.50	102.60	100.58	98.20	95.10
Normal Range of Scores		75–107	73–107	68–101	68–100	60–91

* based on Rikli and Jones [18] (source: own research); ** y.o.—years old.

**Table 6 ijerph-17-01699-t006:** Comparison of the functional fitness levels of older adult men in different age categories with the American standards.

Course of the Trial	Group	60–64 y.o.	65–69 y.o.	70–74 y.o.	75–79 y.o.	80–84 y.o.	≥ 85 y.o.
arm curl	1	18.52	19.68	16.59	17.00	16.10	16.10
	2	19.60	20.50	19.10	18.40	17.40	16.90
Normal Range of Scores *		16–22	15–21	14–21	13–19	13–19	11–17
30 s chair stand	1	11.14	13.12	13.67	13.60	13.50	11.40
	2	12.20	13.80	14.10	14.30	14.00	12.60
Normal Range of Scores		14–19	12–18	12–17	11–17	10–15	8–14
back scratch (cm)	1	−16.16	−15.14	−12.05	−13.40	−13.70	−21.90
	2	−10.40	−12.80	−11.10	−12.40	−13.00	−18.90
Normal Range of Scores		−6.5–0.0	−7.5–1.0	−8.0–1.0	−9.0–−2.0	−9.5–−2.0	−10.0–−3.0
chair sit-and-reach (cm)	1	0.59	2.34	1.35	2.07	6.91	−7.40
	2	1.50	3.60	3.25	3.00	4.10	3.50
Normal Range of Scores		−2.5–4.0	−3.3–3.0	−3.5–2.5	−4.0–2.0	−5.5–1.5	−5.5–0.5
8-foot up and go (s)	1	7.24	7.04	6.49	6.76	7.55	6.36
	2	5.84	6.10	6.50	6.58	7.11	6.24
Normal Range of Scores		5.6–3.8	5.7–4.3	6.0–4.2	7.2–4.6	7.6–5.2	8.9–5.3
2-min step test (s)	1	80.00	82.20	85.60	84.90	75.74	72.10
	2	86.55	84.25	81.34	80.28	78.40	74.90
Normal Range of Scores		87–115	86–116	80–110	73–109	71–103	59–91

* based on Rikli and Jones [18] (source: own research).

**Table 7 ijerph-17-01699-t007:** Pearson correlation coefficient for the functional fitness levels and the age of the older adults tested.

Course of the Trial	Senior Women		Senior Men	
	Group 1	Group 2	Group 1	Group 2
arm curl	−0.032	−0.089	−0.187	−0.189
30 s chair stand	−0.097	−0.027	−0.223 *	0.117
back scratch (cm)	0.051	−0.218 *	−0.231 *	0.219 *
chair sit-and-reach (cm)	0.036	0.067	−0.045	−0.199 *
8-foot up and go (s)	−0.023	0.231 *	0.019	−0.034
2-min step test (s)	−0.046	0.263 *	−0.087	−0.054

* statistically significant difference at α = 0.05 (source: own research).

**Table 8 ijerph-17-01699-t008:** Pearson correlation coefficient for the functional fitness level and BMI of the seniors tested.

Course of the Trial	Senior Women		Senior Men	
	Group 1	Group 2	Group 1	Group 2
arm curl	−0.044	−0.089	−0.095	−0.189
30 s chair stand	−0.092	−0.027	−0.028	0.117
back scratch (cm)	0.051	−0.218 *	0.012	0.219 *
chair sit-and-reach (cm)	0.036	0.067	−0.037	−0.199 *
8-foot up and go (s)	−0.021	0.231 *	0.021	−0.034
2-min step test (s)	−0.048	0.263 *	−0.073	−0.054

* statistically significant difference at α = 0.05 (source: own research).

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
