# Peer review of "Factors Differentiating the Level of Functional Fitness in Polish Seniors"

_ijerph, 2020, doi:10.3390/ijerph17051699_

Round 1
Reviewer 1 Report
The paper sent for review touches upon the issue of measuring the level of functional fitness among seniors, important from the social point of view. The research process has been carried out correctly; however, there is no indication of the practical application of the test results obtained and conclusions drawn.
Detailed comments:
The issue presented in the present work is described in global literature quite extensively. In the paper sent for review, the introduction is based on five reference items, and only two of them contain detailed studies and are available in English. This review does not imply the need for further research. I suggest using a larger number of English-language publications, whose review would necessitate further research on the subject. The statement: "Therefore, due to the accumulated differences in health-related behaviors over a lifetime, this age group is characterized by the greatest diversity in motor skills" seems controversial to me. The title of the paper implies an attempt to evaluate the impact of numerous factors on the level of functional fitness, while the paper itself analyses only selected somatic parameters and the level of physical activity. I suggest that the characteristics of the subjects should be presented in the table. In my opinion, the characteristics of the groups should be more detailed, especially with respect to the level of undertaken physical activity. The subjects of group 2 regularly participated in various forms of physical activity - how often, at what level of intensity? The comparison of the test results obtained in SFT with the standards only on the basis of the juxtaposition of an average group with the standards is incomplete, In my opinion, it would be clearer to present correlation coefficients in charts than in tables, Discussion requires rewording. It should focus on trying to explain the phenomenon to a greater extent than presenting the results of studies done by other authors. In particular, in the case of results deviating from generally accepted regularities, e.g. There was no correlation between the ... age and the level of functional fitness in women on rehabilitation holidays. The conclusions should be more generalized. Conclusion 4 "The level of functional fitness among the tested Polish seniors was higher than that of their American peers" does not result directly from the conducted research.
Author Response
Dear Reviewer,
Thank you very much for the detailed review and for showing us the errors that need to be corrected. We hope you find our corrections satisfying.
Detailed comments:
The issue presented in the present work is described in global literature quite extensively. In the paper sent for review, the introduction is based on five reference items, and only two of them contain detailed studies and are available in English. This review does not imply the need for further research. I suggest using a larger number of English-language publications, whose review would necessitate further research on the subject.
The work was supplemented with additional references.
The statement: "Therefore, due to the accumulated differences in health-related behaviors over a lifetime, this age group is characterized by the greatest diversity in motor skills" seems controversial to me.
The controversial statement was removed from the article.
I suggest that the characteristics of the subjects should be presented in the table.
Table 2 presents a detailed breakdown of the researched group by gender.
In my opinion, the characteristics of the groups should be more detailed, especially with respect to the level of undertaken physical activity. The subjects of group 2 regularly participated in various forms of physical activity - how often, at what level of intensity?
Forms of physical activity in both researched groups were presented.
The comparison of the test results obtained in SFT with the standards only on the basis of the juxtaposition of an average group with the standards is incomplete, In my opinion, it would be clearer to present correlation coefficients in charts than in tables.
Due to the need to place 24 values, the chart was abandoned and presented in the table.
Discussion requires rewording. It should focus on trying to explain the phenomenon to a greater extent than presenting the results of studies done by other authors. In particular, in the case of results deviating from generally accepted regularities, e.g. There was no correlation between the ... age and the level of functional fitness in women on rehabilitation holidays.
The discussion has been rewritten.
The conclusions should be more generalized.
As suggested by the reviewer, the conclusions have been generalized. Conclusion 4 "The level of functional fitness among the tested Polish seniors was higher than that of their American peers" does not result directly from the conducted research.
We gave up this conclusion and moved this information to discuss the results.

Reviewer 2 Report
The analysis of functional capacity in the elderly is at its peak, especially after the definition of WHO in 2001 regarding active aging.
The study is interesting but some factors are missing to clarify the reader. My assessment will be global in some points and specify in others.
The materials and methods are confusing. Only in the results do we find out exactly which variables will be tested (arm curl, chair sit and reach and 8-foot up and go, back scratch test, 30 second chair stand and 2-Minute StepTest), should be defined in the methodology.
This section already presents results, such as height, weight and BMI, these must be at the beginning of the results.
It presents two groups, how were they separated? Just by integrating one or the other site (Factor that in itself can already define the difference in results between groups!). So, were they conveniently separated?
Another factor is the difference in the n sample between groups, it can balance the groups with the same n sample. Another aspect that is not clear is how many elements there are in each group, it only indicates the difference between men and women but it does not indicate how much there are in group 1 and group 2!
Something that would be important to explore is what is practiced at the camp? Any physical activity? We compare things that are comparable, I can have the homogeneity analysis by age, weight, gender, BMI, etc. (which is also not clear through which variable was made) but, it does not mean that the groups are homogeneous, they may have characteristics physical and biopsychosocial complement, different!
One more detail would be to check the before and after of a training period with group 2 and compare it with group 1 in the same period (pre and post test). To check the real difference between groups!
In the results, the test is called t-Student and not t-Studenta
In tables 1 and 2, it would be important to demonstrate the p-value of the difference between groups and to verify the existence or not of statistical differences.
After a good discussion, a good descriptive conclusion is expected, and not by topics as presented, which does not match the structure of the journal.
This theme has already been addressed several times in the journal, where presenting only one reference does not represent the real interest in the works published by the journal.
I realize how hard it is to carry out such a study, but I recommend a review before it is approved.
Author Response
Dear Reviewer,
Thank you very much for the detailed review and for showing us the errors that need to be corrected. We hope you find our corrections satisfying.
Detailed comments:
The materials and methods are confusing. Only in the results do we find out exactly which variables will be tested (arm curl, chair sit and reach and 8-foot up and go, back scratch test, 30 second chair stand and 2-Minute StepTest), should be defined in the methodology.
Test attempts’ names were supplemented in the methodology and placed in table 1.
This section already presents results, such as height, weight and BMI, these must be at the beginning of the results.
The results regarding the somatic parameters were moved from ‘Material and methods’ chapter to ‘Results’ chapter.
It presents two groups, how were they separated? Just by integrating one or the other site (Factor that in itself can already define the difference in results between groups!). So, were they conveniently separated?
The number of respondents in groups 1 and 2 was supplemented and detailed characteristics of both groups were given. It was explained how the division into group 1 and group 2 was made.
Another factor is the difference in the n sample between groups, it can balance the groups with the same n sample. Another aspect that is not clear is how many elements there are in each group, it only indicates the difference between men and women but it does not indicate how much there are in group 1 and group 2!
Table 2 shows the division of the researched group into groups 1 and 2, taking into account the sex of the respondents.
In the results, the test is called t-Student and not t-Studenta.
The error in the name t-Student has been corrected.
In tables 1 and 2, it would be important to demonstrate the p-value of the difference between groups and to verify the existence or not of statistical differences.
In tables 1 and 2 in the column with the name ‘d’ the difference between the samples in groups 1 and 2 is given, and in the last column the statistical significance of the differences at the level of p = 0.001 is given and marked **.
After a good discussion, a good descriptive conclusion is expected, and not by topics as presented, which does not match the structure of the journal.
The discussion has been rewritten. The work was supplemented with additional references. The conlusions were reworded.

Round 2
Reviewer 2 Report
Dear authors,
The study is interesting but some aspects still need clarification and proposed improvements. My assessment will be global in some points and specify in others.
The difference in the n sample between groups, it can balance the groups with the same n sample. for example, each group has 105 men and 153 women! Or simply level the groups equally, each group having 210 people (105 men and 105 best). allowing linearity of groups.
The initial analysis must take into account the homogeneity test according to Pearson. Regarding age and BMI for example.
One more detail would be to check the before and after of a training period with group 2 and compare it with group 1 in the same period (pre and post test). To check the real difference between groups! This analysis must also be done globally (Group 1 VS Group 2) and not only the comparison between men and women, which is also important
In the results, the test is called t-Student and not t-Studenta (Changed in text but not changed in tables).
I find the analysis by age group interesting, but are there differences between each group before and after the exercise program in each age group? Or does it only occur globally with the average age?
I realize how hard it is to carry out such a study, and I know how difficult it is to move the data after it is done, but these tips can enhance the validity of the data and its homogeneity, but I recommend a review before it is approved.
Author Response
Detailed comments:
The difference in the n sample between groups, it can balance the groups with the same n sample. for example, each group has 105 men and 153 women! Or simply level the groups equally, each group having 210 people (105 men and 105 best). allowing linearity of groups.
The numbers in the research groups were reduced as suggested by the reviewer - for men the number of 105 people and for women 153 people.
The initial analysis must take into account the homogeneity test according to Pearson. Regarding age and BMI for example.
In lines 106-109 contain information about assessing the normality of the distribution of the studied variables using the Shapiro-Wilk test.
One more detail would be to check the before and after of a training period with group 2 and compare it with group 1 in the same period (pre and post test). To check the real difference between groups! This analysis must also be done globally (Group 1 VS Group 2) and not only the comparison between men and women, which is also important
The aim of the research was to show the level of functional fitness of seniors, and not to analysis for changes that occurred under the influence of conducted exercises (in group 1 and in group 2). Therefore, only one measurement was analysed. Tables 3 and 4 show the differences between group 1 and group 2 for women and men. In contrast, the work did not compare the level of fitness between men and women, because there are no point tables for converting the results from individual trials to points. The existence of such tables would allow comparison of Fullerton test samples regardless of the sex of the respondents and the type of test performed.
In the results, the test is called t-Student and not t-Studenta (Changed in text but not changed in tables).
The error in the name t-Student in tables 3 and 4 has been corrected.
I find the analysis by age group interesting, but are there differences between each group before and after the exercise program in each age group? Or does it only occur globally with the average age?
The presented work did not measure changes in groups before and after the program. Therefore, this problem could not be presented in terms of age category.
